# Mesenchymal Stem Cell-Secreted Exosomes and Soluble Signals Regulate Breast Cancer Metastatic Dormancy: Current Progress and Future Outlook

**DOI:** 10.3390/ijms25137133

**Published:** 2024-06-28

**Authors:** Bei Dai, Amanda M. Clark, Alan Wells

**Affiliations:** 1Department of Pathology, School of Medicine, University of Pittsburgh, Pittsburgh, PA 15213, USA; daibei@pitt.edu (B.D.); amc235@pitt.edu (A.M.C.); 2R&D Service, Pittsburgh VA Health System, Pittsburgh, PA 15213, USA; 3School of Medicine, Tsinghua University, Beijing 100084, China; 4Cell Biology Program, Hillman Cancer Center, University of Pittsburgh, Pittsburgh, PA 15213, USA; 5Pittsburgh Liver Research Center, University of Pittsburgh, Pittsburgh, PA 15213, USA

**Keywords:** metastatic dormancy, mesenchymal stem cells, secretome, metastatic breast cancer

## Abstract

Breast cancer is most common in women, and in most cases there is no evidence of spread and the primary tumor is removed, resulting in a ‘cure’. However, in 10% to 30% of these women, distant metastases recur after years to decades. This is due to breast cancer cells disseminating to distant organs and lying quiescent. This is called metastatic dormancy. Dormant cells are generally resistant to chemotherapy, hormone therapy and immunotherapy as they are non-cycling and receive survival signals from their microenvironment. In this state, they are clinically irrelevant. However, risk factors, including aging and inflammation can awaken dormant cells and cause breast cancer recurrences, which may happen even more than ten years after the primary tumor removal. How these breast cancer cells remain in dormancy is being unraveled. A key element appears to be the mesenchymal stem cells in the bone marrow that have been shown to promote breast cancer metastatic dormancy in recent studies. Indirect co-culture, direct co-culture and exosome extraction were conducted to investigate the modes of signal operation. Multiple signaling molecules act in this process including both protein factors and microRNAs. We integrate these studies to summarize current findings and gaps in the field and suggest future research directions for this field.

## 1. Introduction

### 1.1. Metastasis as the Key Mortal Step in Breast Cancer

Breast cancer is one of the most common cancers worldwide and the leading cause of cancer deaths in women [1]. In recent years, breast cancer accounts for around 30% new female cancer cases and 15% of new deaths of female cancer in the United States, making it the most common cancer type with the second highest death toll for American women [2,3,4]. Despite this high death toll, most breast cancers are caught early when there is no evidence of distant spread; these cancers are treated as localized diseases and removed. When these breast-limited cancers are removed the 10-year survival rate exceeds 80% [5]. Yet, many of these seemingly ‘cured’ cancers recur as metastases in distant sites, leading to mortality years to decades later. Thus, understanding why the disseminated breast cancer cells (BCC) reactivate and grow into lethal tumors is critical to conquering this cancer.

At least part of the predilection for recurrence is hardwired in the BCC themselves. The cells that lead to these cancers express female sex hormone receptors for estrogen (ER) and progesterone (PR) and are responsive to these hormones; however, as some cancers develop, they become resistant to signaling from ER and PR. This is one of the major divides in breast cancer classification (in addition to the proposed cell of origin of the breast cancer), decided by the “positive or negative” state of ER, PR, as well as a tumor growth promoting receptor called human epidermal growth factor receptor 2 (Her2). The distinction of ER expression not only determines treatment, with ER-positive tumors being subjected to hormonal blockade, but also prognosis. For ER-negative patients, which account for 20–40% of all breast cancers [6], the distant recurrence usually occurs 3–5 years after primary tumor diagnosis; while for ER-positive patients, most breast cancer recurrence appears 5 years after the diagnosis and remains possible even 15 years later [6,7,8]. Within 20 years following diagnosis, 13–41% of ER-positive patients with endocrine therapy for 5 years will experience tumor recurrence, mainly leading to deaths [7]. Because of the possibility of recurrence, breast cancer is considered as an “incurable” disease.

### 1.2. Metastatic Dormancy and Emergence

To explain this observation that tumors can seem absent for years before emerging often at sites distant from the primary tumor (Figure 1), Rupert Willis proposed a concept called “tumor dormancy” in 1934 [9], and this concept was gradually developed and expanded by later generations. Briefly, disseminated tumor cells (DTCs) lay dormant and quiescent for years before they ‘awaken’ as aggressive tumor cells. The dormancy at a secondary site rather than the primary site is called “metastatic dormancy” [8], which will be discussed in this perspective. Dormancy can also be divided into tumor mass dormancy and cellular dormancy. The former indicates an equilibrium of small tumors with approximately constant mass, while in the latter, most single tumor cells or clusters are growth-arrested in G_0_/G_1_ phases [10,11,12]. During the dormant period, DTCs are non-cycling most of the time and receive survival signals from the environment, making them resistant to chemotherapy [13,14], hormone therapy, and immunotherapy [15,16,17]. Adjuvant endocrine therapy for ER-positive breast cancers and chemotherapy for triple negative breast cancer (TNBC) cancer can only reduce the local and distant recurrence risks and mortality by about one-third or less [7,18]. Thus, understanding dormancy is critical to improving the long-term survival of breast cancer.

Factors inducing dormancy or reactivation that are not restricted to mesenchymal stem cells (MSCs) are complex and are still being explored. It has been demonstrated that the host ectopic tissue can impose dormancy [19,20]; and this dormancy coincides with the reversion of aggressive DTCs to a more epithelial state [14,19,21]. While the specific signals that impose dormancy in the course of spontaneous metastasis are still being deciphered, a number of events can drive cellular or tumor mass dormancy. Briefly, these include multiple physical factors (including local energy minima, mechanical confinement, and hypoxia) [22,23], cellular factors (including mesenchymal stem cells, fibroblast, and T cells), structural protein (including fibronectin), and soluble factors (including BMP4, BMP7, TGF-β1 and β2, and FGF-2) [24,25].

Dormancy is considered as a meta-stable state given the mutational burdens of BCCs that predispose to uncontrolled proliferation. There are many hypotheses of the ‘awakening’ signals, and it is a still unanswered question if micro-metastases can undergo cycles of dormancy and emergence until the dormancy-inducing signals fail to overcome the reawakening risk factors. Therefore, equally important to understand are the events that trigger the emergence from dormancy or the ‘awakening’ of the DTCs, as this leads to lethal metastatic outgrowths. Again, while these are being deciphered in humans, there are a number of candidates that have been shown in experimental model systems. Signals released during tissue inflammation, such as growth factors (e.g., EGF) [26], cytokines (e.g., IL8/CXCL8) [27], and chemokines (e.g., CXCL10) [28], drive emergence in ex vivo models of dormancy; and chronic stressors such as major surgeries and myocardial infarction are related to subsequent emergence in humans [29,30]. Physical aspects of the tissues can also drive mesenchymal reversion and awakening [31,32]; as well as the cellular make-up of the metastatic organ including tissue macrophages and adiposity [33,34]. Whether these signals derive locally or distantly remains to be determined. However, the cellular origin of such signals is critical for rational interventions.

**Figure 1 ijms-25-07133-f001:**
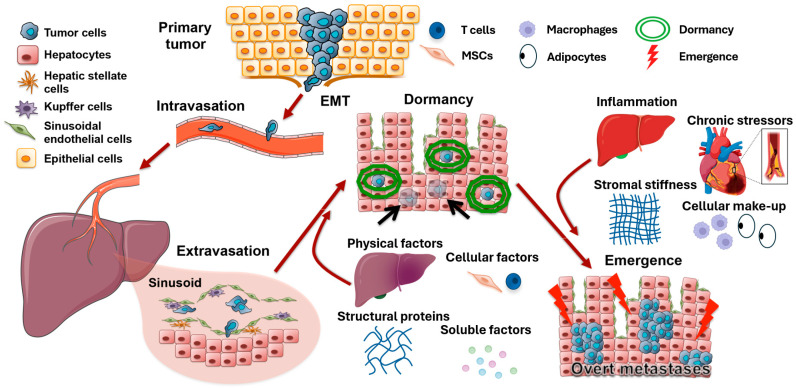
Tumor metastatic cascade and the risk factors of dormancy and emergence. After undergoing epithelial–mesenchymal transition, tumor cells enter the bloodstream, migrate to metastatic sites (represented by liver), and exit the vessels upon arrival. Physical factors such as hypoxia, influences from other cells (such as MSCs and T cells), structural proteins, and various soluble factors then induce tumor cell dormancy, making tumor cells evade initial tumor treatments. Even many years later, inflammation, chronic stress, stromal stiffness, and the cellular make-up of the metastatic organ can reactivate dormant tumor cells, leading to recurrence. Modified from Clark et al. (2016) [35].

### 1.3. Mesenchymal Stem Cells

DTCs are found in many organs, including such diverse tissues as the liver, lungs, bone marrow, and brain; and in all these tissues, the DTCs can be seen in a state of dormancy, and emerge later as lethal outgrowths. As dormancy appears to be under at least partial regulation by the local microenvironment, there is a search for commonality across these tissues of distinct ontologies. Thus, in addition to immune cells, it provides the initial impulse to look at the influence on breast cancer progression and dormancy of MSCs that can be both tissue-resident and from circulating bone marrow-derived stem cells.

Tissue-resident MSCs can be isolated from various tissues such as brain, spleen, liver, lung, bone marrow, muscle [36], and fetal origins including umbilical cord [37] and placenta [38]. MSCs extensively participate in tissue regeneration and maintenance due to their differentiation ability, while they are also anti-inflammatory and have immunomodulatory effects that suppress the proliferation and function of immune cells such as T cells, B cells, and natural killer cells [37] as well as promote the differentiation of macrophages to anti-inflammatory phenotype M2 [39]. As chronic inflammation is one of the events that trigger emergence, the immunomodulation is consistent with the dormancy inducing effects that will be introduced in the perspective. Among different sources, bone marrow-derived MSCs (BMMSCs) are applied in cell therapy more frequently [40]. BMMSCs will migrate to injured tissues through the circulatory system and join the healing process via the homing effect mediated by a variety of cytokines, chemokines, and growth factors [37], among which a well-known one is SDF-1/CXCR4 signaling pathway [41]. Thus, effects on DTC can be from both tissue resident and circulating MSC.

### 1.4. MSCs in Metastatic Breast Cancer

Inflammation will recruit MSCs, including the local inflammation created by tumor cells. As a result, MSCs migrate to the tumor area, communicate with tumor cells, and become an essential part of the tumor microenvironment [42], whose interactions with tumor cells influence the whole process from tumor initiation, progression, and metastasis to the response to therapies [43,44]. MSCs have not only been shown to be recruited to the primary tumor site by soluble molecules secreted by BCC [45], but also migrate with BCCs from the primary tumor to facilitate distant metastasis in animal models [46] and can be isolated from human breast cancer metastasis [46,47], implying they may have effects on breast cancer.

These studies thus posit a role for MSCs in breast cancer metastasis, dormancy, and progression. In the following, we look at recent findings in how the intercell signaling regulates BCC fates.

## 2. Signaling Mechanisms

### 2.1. Juxtacrine and Near-Paracrine Signaling

Most of the in vitro findings suggest that MSCs promote metastatic dormancy with diverse effects on proliferation and cancer cells properties based on both direct or indirect co-culture of the MSCs with the BCCs, as illustrated in Figure 2a.

Both Transwell co-culture [48] with adipose MSCs and direct co-culture [49] with MSCs result in G_0_/G_1_ arrest of BCCs. However, one can find contradictory results about MSCs’ stimulatory effects on BCC proliferation in Transwell co-culture with adipose MSCs [48,50], direct co-culture with MSCs in vitro [51,52] or after treatment of MSC conditioned media [53]. This may suggest that the effects of MSC on breast cancer cell proliferation are context dependent, which is also suggested by opposite correlations in the primary and metastatic tumor sites, with an example following. Exposure to NG2^+^/Nestin^+^ MSCs in the bone marrow results in reduced levels of proliferation marker (Ki67 and PH3) expression and increased autophagy marker (p-ATP2 and p27) expression [54]. However, this was not observed in the direct and indirect co-culture experiments in vitro (EdU as a proliferation marker; LC3B as an autophagy marker) [55] and it is also shown that co-injected MSCs stimulate BCC proliferation in the primary xenograft in vivo [56,57,58].

CD44, as a marker for cancer stem cells, was downregulated in BCCs co-cultured with MSCs; this led to a decrease in proliferation and an increase in chemo-resistance (represented by reduced killing by docetaxel) [51], which is concordant as rapidly proliferating cells are usually more susceptible to chemotherapy, whereas there is also evidence suggesting the promotion of MSCs on breast cancer stem cell phenotypes and tumor progression [59,60].

Additionally, Transwell co-culture with adipose MSCs also inhibits the migration and invasion abilities of BCCs, triggers the transition from mesenchymal markers to epithelial markers (i.e., decreases in vimentin, ZEB2, SMAD4 and increases in E-cadherin etc.), reduces sensitivity to multiple chemotherapy drugs (doxorubicin, cisplatin, and 5HNQ), and dysregulates genes involved in drug resistance, cancer stem cell and DNA repair [48]. MSC direct or indirect co-culture do not impact the viability of BCCs, but the indirect co-culture stimulates their metabolic activity [55].

In brief, experimental results in published works may seem contradictory at first glance. However, the MSC effects are likely contextual with promotion of progression most often seen in the primary tumor setting, while we see suppressive events driven by MSCs in the setting of early establishment in the metastatic setting. This is likely due to the complex interactions of the diverse cell types and metabolic challenges that impact the tumor cell phenotypes. However, more complete discussion of these networks awaits more research and lies beyond the scope of this targeted perspective.

### 2.2. Exosomes Are Signaling Elements

Recent findings have shown that exosomes are important in signaling between BCCs and the host metastatic environment [61,62]. This is critical as exosomes transfer not just proteins for temporary signaling similar to soluble signals as above, but also microRNAs to reprogram the transcriptional and transcribed profiles of the BCCs. The MSC exosomes production is of interest in examining the more persistent changes in the BCCs. Of interest, the exosome composition changes dependent on bi-directional communication between the MSCs and BCCs. MSCs indirectly stimulated by BCCs secrete “primed exosomes” while MSCs alone secrete “naïve exosomes”; it is important to note that the cargo differs between primed and naïve exosomes and thus may exert different effects. Both primed and naïve exosome treatment can transition BCCs into G_0_/G_1_ arrest [63]. But it was also observed that naïve exosomes can enhance BCC fractions in cycling phases [64]. BCCs treated with primed or naïve exosomes exhibit reduced growth potential [48,51,63,65], stronger autophagy [63], weaker metabolic activity (including reactive oxygen species, mitochondrial superoxide, and manganese superoxide dismutase) [63], cancer stem cell transition [63,65], and doxorubicin resistance [63]. Apart from that, naïve exosomes can also suppress the invasiveness and the migration ability of BCCs [48], enhance their cell adhesion [65], and reversibly restrain their CD44 (cancer stem cell marker) level [51]. These alterations may explain why in the metastatic niche the MSCs are predisposed to inducing dormancy as they produce exosomes that are distinct from tissue resident MSC in the primary tumor (parenthetically, this leads to testable hypotheses for future studies). Although all studies about programmed cell death caused by MSC exosomes are restricted to autophagy, it is shown that MSC derived micro-vesicles induce apoptosis of BCCs on polycaprolactone nanofibers [66]. While these studies suggest a role for exosomes in imposing a less proliferative and aggressive phenotype on BCCs, indicative of dormancy, summarized in Figure 2b, the large and diverse nature of the cargo of exosomes leaves open many questions as to the operative signaling mechanisms.

## 3. MSC-Involved Cell Fusion as a Signaling Trigger

In addition to the signaling described above, a novel discovery has emerged in recent years that BCCs may fuse with MSCs when in contact, and this could lead to both phenotype switches and novel signaling [67,68,69]. The cell fusion between BCCs and MSCs has been reported to result in multiple effects on the tumorigenesis and aggressiveness of the BCCs. Different subtypes were identified of these hybrid cells when fusing aggressive MDA-MB-231 TNBC cells with MSCs in vitro. Some subtypes (represented hyb1 and hyb2) are more aggressive both in vivo and in vitro [67], while others (represented by hyb3 and hyb4) display weaker or the same proliferative capacity and tumorigenicity [68]. In addition to these two kinds, one specific subtype (hyb5) is different as it spontaneously entered dormancy after injection into mouse models. The original incidence of this hyb5 tumors was delayed for up to half a year compared with the parental BCCs, but the tumors grew faster and metastasized more once reawakened (Figure 2c); such dynamics are often seen in late recurrences in patients. In vitro culture results are not only consistent with the in vivo results in terms of enhancing proliferation and altering expression of dormancy-associated genes, but also demonstrate the hyb5 subtype is more sensitive to chemotherapy drugs represented by taxol and cyclophosphamide [69]. However, currently, there are only a limited number of studies concerning cell fusion between BCCs and MSCs. In these studies, there is no means to verify the fusion state leading to uncertainty of how cancer cells are affected throughout the process, which should be the topic of future studies.

Hybrid cells only account for 0.1–2% of the total cell populations of BCCs and MSCs [70], indicating their number is also very small in the actual disease. The number of subtypes that spontaneously enter dormancy would be even smaller as different subtypes with diverse phenotypes are generated during the cell fusion. Still, as metastatic seeding is a rare event on the order of <1% of disseminated cells [71], this may contribute to initial dormancy in the metastatic target organ. Additionally, TNF-α, a kind of pro-inflammatory cytokine, was shown to enhance these cell fusions via MAPK8, NF-κB or cell death pathways [70], which harken back to the non-specific foreign body response being active during tumor dissemination and leading to the cell fusion with MSCs.

## 4. Signaling Pathways

Understanding the molecular mechanism that is triggered by these signals is the first step in designing rational approaches to the question of metastases. Though only a limited number of studies have been reported investigating the signaling pathways involved in the MSC-promoted dormancy of BCCs, a model can still be proposed. In the metastatic sites, MSCs facilitate dormancy of BCCs by not only secreting signaling molecules including TGF-β2, BMP7, and SDF-1α through paracrine signaling, but also some of the miRNAs in the MSC-secreted exosomes including miRNA-941, miR-23b, and miR-222/223. As direct co-culture is not necessary to induce BCC dormancy, it is still not clear if any juxtacrine signaling is involved in the MSC-promoted BCC dormancy.

NG2^+^/Nestin^+^ MSC-secreted TGF-β2 and BMP7 were identified as dormancy inducers in the bone marrow via receptors TGFBRIII and BMPRII respectively with the intermediary signaling molecules p38, p27, and ERK in the downstream pathways. More specifically, abrogation of TGF-β2 and BMP7 in vivo lead to greater metastatic load and outgrowth concomitant with lower levels of BCC dormant markers. To further support this as key pathways, analysis of breast cancer patients without systemic recurrence show higher frequency of TGF-β2 and BMP7, while the disease-free survival rate of patients with detectable BMP7 is higher than those without BMP7 [54].

BCC-expressed NK1R-Tr is another dormancy-associated signal molecule. After co-cultured with MSCs, NK1R-Tr expression levels in BCCs were generally lower (decreased then increased, but still lower than the beginning) [49]. NK1R-Tr knockdown of BCCs slows down their growth speed significantly and NK1R-Tr overexpression protects breast cancer cell line HBL-100 from the co-culture with MSCs, where parental HBL-100 cells die. SDF-1α, which is mainly released by MSCs, restricts NK1R-Tr expression in BCCs, while TGF-β1, which is released by both BCCs and MSCs, facilitates NK1R-Tr expression in BCCs. The proliferation inhibition caused by MSC co-culture can be rescued by TGF-β1 knockdown in BCCs, indicating TGF-β1 expressed by BCCs works in this process. However, it cannot be rescued by SDF-1α-neutralizing antibody, which may imply a more complicated signal transmission mode related to the SDF-1α outside of BCCs [49]. Therefore, this inconformity implies the relationship between MSC-secreted SDF-1α and TGF-β1 and BCC dormancy is relatively unclear, where further research is needed.

Given the signaling from exosomes, there has been a focus on the microRNAs transferred to BCCs. By comparing the co-culture conditioned media and MSC conditioned media or comparing the primed exosomes and naïve exosomes, miRNA-941 [48], miR-23b [51], and miR-222/223 [64] were found to be candidates. MiRNA-941 impedes the proliferation, invasion, migration of BCCs and triggers the marker transition from mesenchymal to epithelial [48]. MiR-23b which targets MARCKS, inhibits the proliferation capacity and CD44 (cancer stem cell marker) expression of BCCs [51]. MiR-222/223 activates P-glycoprotein and inhibits drug sensitivity and proliferation of BCCs [64]. Combination therapy of anti-miR-222/223 and carboplatin shows better survival rates compared with the “control anti-miRNA + carboplatin” group in a mouse model of breast cancer [64].

## 5. Conclusions

In conclusion, recent studies consistently suggest MSCs co-culture, MSC-secreted exosomes, or cell fusion promotes the dormancy of BCCs defined by the phenotype that includes reduced proliferation, enhanced autophagy, resistance to chemotherapy, and similarity with cancer stem cells and epithelial cells. However, many of these investigations are in a relatively early stage of research. Proteins or miRNAs were proposed as candidates even with supporting evidence of downstream signaling pathways.

Diametrically opposite opportunities are presented for translating metastatic dormancy to clinical therapies. Deciphering the mechanisms by which BCCs enter into or emerge from dormancy is the cornerstone of research into treatments. One could attempt to prolong the dormancy as long as possible, or contrapointally to reawaken the tumor cells combined with conventional cancer treatment such as chemotherapy that is possibly a prototype of an innovative therapy. The only therapy-related animal experiment conducted in the 9 studies discussed in this perspective is the latter. The regimen group of anti-miR-222/223 (to awaken dormant cells) and carboplatin (a chemotherapy drug) had higher survival rates compared with those treated with control anti-miRNA [64]. However, the differences between mice and humans in terms of cancer growth and timescales cannot be ignored, either. If there is a risk of deterioration when tumor cells are reawakened but not killed, it might even be better to leave them dormant and wait for the delayed recurrence.

Future studies will need to delineate a clearer mechanism by which MSCs regulate breast cancer metastatic dormancy. More signal molecules, their correlations, and related signaling pathways are to be investigated. Depending on whether we find a signal that is sufficient to induce dormancy on its own, or those required to maintain dormancy will dictate which of these two therapeutic approaches will dominate.

## Figures and Tables

**Figure 2 ijms-25-07133-f002:**
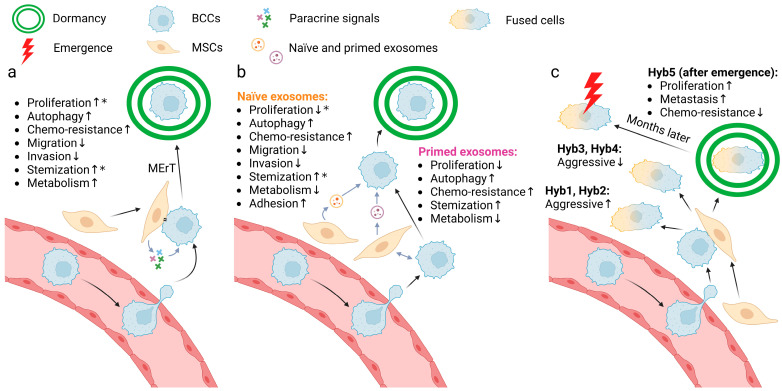
Three models of the mechanism of MSC-induced dormancy of BCCs. (**a**) Juxtacrine and paracrine signals from MSCs guide BCC into dormancy, characterized by several phenotypes from weaker proliferation to stronger metabolism. *, conflicting results exist in current research. (**b**) “Primed exosomes” and “naïve exosomes” induce BCC dormancy, characterized by different phenotypes. *, conflicting results exist in current research. (**c**) Cell fusion of MSCs and BCCs yields multiple subtypes with different characteristics. Among them, hyb5 enters dormancy spontaneously and becomes reactivated after months. The arrows after the processes indicate the direction of the effect of the signaling. Created with BioRender.com (accessed on 10 May 2024).

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
