# Peer review of "Mesenchymal Stem Cell-Secreted Exosomes and Soluble Signals Regulate Breast Cancer Metastatic Dormancy: Current Progress and Future Outlook"

_ijms, 2024, doi:10.3390/ijms25137133_

Round 1

Reviewer 1 Report

Comments and Suggestions for Authors

Dai and coworkers highlight in this review an important point about the related dormancy of breast cancer cells and MSCs. Some improvements can be made to better understand the review, as shown below:

A) Please review whether the form of citation in the text using Roman numerals is indeed usual and the recommendation for this journal. The use of Arabic numerals makes it easier to find the reference.

B) The inclusion of the sentence in parentheses between lines 54 and 56 makes it difficult to understand the sentence. I suggest removing it to another location in the text.

C) The beginning of the sentence on line 80 is broken. I suggest including: "It has been shown that.."

D) Cite the figures in the text before presenting them.

E) The figure legend does not meet the formatting standard required by the journal.

F) Line 116: Change "the most.." to "more..".

G) Line 125: the correct is "..progression, and metastasis..."

H) Apparently there is a formatting error on line 134.

I) From the session "Juxtacrine and near-paracrine signaling" onwards, there are a series of bibliographic citation errors in the text.

J) It was not clear how the decrease in the expression of a marker related to proliferation led to chemoresistance by BCC, shouldn't it be the opposite? Were the same cells that were co-cultured with MSCs also subsequently exposed to docetaxel?

K) There must be confusion between lines 149 and 152, because if MSCs inhibit the invasive and migratory capacity of BCC, how do they then promote the mesenchymal to epithelial transition?

L) Are there no results regarding the apoptotic potential of MSC exosomes on BCC?

M) The authors could conclude the work by including a reflection on this dualistic role that MSCs play, both inhibiting the proliferative capacity of BCCs and also participating in their dormancy. Furthermore, they must highlight the importance of research that clarifies the mechanisms by which BCCs abandon dormancy.

N) Please review all manuscript formatting.

Comments on the Quality of English Language

The English of the manuscript needs a complete revision, as in some parts the reading is confusing.

Reviewer 2 Report

Comments and Suggestions for Authors

The formation of exosomes and their role in metastasis formation is a really interesting field. I miss some information about matrix metalloproteinases in this process.

There are some labelled errors (mostly citations are missing) in the manuscript, please correct them.

Reviewer 3 Report

Comments and Suggestions for Authors

Major critiques:

1)     The main point of the review is that mesenchymal stem cells (MSC) induce and control dormancy of breast cancer (BC) metastasis. However, this conclusion does not come across as a very convincing for multiple reasons. (A) As stated in lines 121-133 and well established in the literature, MSC promote tumors, not suppress them. This directly contradicts the notion that MSC inhibit proliferation, migration, and invasion of co-cultured primary tumor cells in vitro or metastatic cells in vivo. (B) The main function of MSC as any other type of stem/progenitors is to regenerate and expand tissue structures (e.g., in wound healing). Once again, this is not consistent with their induction of tumor cell dormancy. (C) MSC are present in every organ for the duration of organism lifespan. It is difficult to explain why these cells would first induce dormancy for 10 years but eventually cease of doing so. (D) The signaling molecules by which MSC presumably induce BC cell dormancy (lines 79-100) have no specificity to this process. For instance, fibronectin is present at high concentration in normal serum whereas mentioned cytokines (e.g., TGFb1) circulate in the blood of healthy subjects albeit at somewhat reduced level than in tumor-bearing mice or humans. None of these molecules or their concentrations are MSC-dependent. The mechanism by which these rather common molecules could suppress mitosis of metastatic cells is not explained. (E) Lines 113-115 indicate that MSC are immunosuppressive. Immunosuppression is a key mechanism that allows tumor cell escape from anti-tumor immune control. If so, why dormant lesions are not growing in the absence of anti-tumor response mediated by MSC-imposed immunosuppressive environment? This MSC property needs to be reconciled with the proposed inhibitory effect on tumor cell behavior.

In short, the presented concept is less than compelling either because the supportive evidence is very superficially described (no numerical assessments or logical mechanisms to explain the MSC effects) or lack of acknowledgement and fair discussion of any conflicting or contradictory evidence, ample amount of which is present in the literature.

2)     Related to the point#1, the supportive arguments for any statement/hypothesis must be backed by logical mechanistic evidence or at least possible explanation. In this article, there is no evidence-based or hypothetical but plausible mechanistic explanations for any of the listed signaling molecules or processes (e.g., fusion) that impose MSC-mediated metastatic dormancy. Simple listing of all reported molecules and observations in individual studies does not make strong arguments.  The authors need critically assess which studies present most compelling cause-effect type of evidence that are corroborated by independent approaches and by multiple research groups.

3)     Lines 102-109 imply that because dormant metastases can be found in organs that also contain MSC, the latter are involved in regulation of dormancy. This is a very poor argument. All mentioned organs are also vascularized and innervated but that does not mean that endothelial or neuronal cells control metastatic cell dormancy.

4)     The concept of MSC-derived exosomes driving BC dormancy suffers from all the deficiencies mentioned above. Exosomes are produced by a variety of different cells so it is not possible to determine in vivo (unless specifically labeled) which cells within the local environment produced critical vesicles. Heterogeneity of the effects also depends on specific exosomal cargo in each context. Exosomes from MSC or other cells have been implicated in both promoting and restricting cell division and other functions, possibly depending on exosome cargo and tissue contextual details.  The information provided here does not contain any definitive or convincing evidence that exosomes from MSC are cell division inhibitory and/or play a major role in metastatic dormancy. Quantitative evidence indicative of a major impact or data derived from multiple independent approaches would provide better support.  

Minor critiques:

 1)     Listing of references is unacceptable. Some references are listed using atypical Roman numbers instead of standard Arabic numericals while many are not listed at all being replaced by Error! sign. The majority pf references cannot be verified due to this negligence in article formatting.

2)     Lines 47-60 – classification of BC subtypes requires editing; it is too lengthy but does not mention basal myoepithelial cells that give rise to most metastatic tumors. 

3)     BC metastases detected after long remission can be in various organs, not only in the bone. The statement that 70% of BC metastases are in the bones is not supported by a reference.

4)     MSC mediated fusion paragraph requires revision. The paragraph seems to describe a single study, and largely ignores substantial body of evidence indicating tumor-promoting and pro-metastatic fate of MSC-tumor cell hybrids.

5)     CD44 is not a proliferative marker (line 147). It is constitutively expressed in many progenitors, tumor cells and blood vascular cells.

Round 2

Reviewer 1 Report

Comments and Suggestions for Authors

The authors provided a significant improvement in the manuscript. All issues were answered accordingly.

Reviewer 3 Report

Comments and Suggestions for Authors

No further comments